# Scion-to-Rootstock Mobile Transcription Factor *CmHY5* Positively Modulates the Nitrate Uptake Capacity of Melon Scion Grafted on Squash Rootstock

**DOI:** 10.3390/ijms24010162

**Published:** 2022-12-22

**Authors:** Shu’an Hou, Yulei Zhu, Xiaofang Wu, Ying Xin, Jieying Guo, Fang Wu, Hanqi Yu, Ziqing Sun, Chuanqiang Xu

**Affiliations:** 1College of Horticulture, Shenyang Agricultural University, Shenyang 110866, China; 2Key Laboratory of Protected Horticulture (Ministry of Education), Shenyang 110866, China; 3Modern Protected Horticultural Engineering & Technology Center, Shenyang 110866, China; 4Key Laboratory of Horticultural Equipment (Ministry of Agriculture and Rural Affairs), Shenyang 110866, China

**Keywords:** rootstock-scion interaction, graft, mobile mRNA, ELONGATED HYPOCOTYL 5 (HY5), nitrate transporter, nitrate uptake

## Abstract

It is generally recognized that the root uptake capacity of grafted plants strongly depends on the rootstocks’ well-developed root system. However, we found that grafted plants showed different nitrate uptake capacities when different varieties of oriental melon scion were grafted onto the same squash rootstock, suggesting that the scion regulated the nitrate uptake capacity of the rootstock root. In this study, we estimated the nitrate uptake capacity of grafted plants with the different oriental melon varieties’ seedlings grafted onto the same squash rootstocks. The results indicated a significant difference in the nitrate uptake rate and activity of two heterologous grafting plants. We also showed a significant difference in *CmoNRT2.1* expression in the roots of two grafting combinations and verified the positive regulation of nitrate uptake by *CmoNRT2.1* expression. In addition, the two varieties of oriental melon scion had highly significant differences in *CmHY5* expression, which was transported to the rootstock and positively induced *CmoHY5-1* and *CmoHY5-2* expression in the rootstock roots. Meanwhile, *CmHY5* could positively regulate *CmoNRT2.1* expression in the rootstock roots. Furthermore, *CmoHY5-1* and *CmoHY5-2* also positively regulated *CmoNRT2.1* expression, respectively, and *CmoHY5-1* dominated the positive regulation of *CmoNRT2.1*, while *CmHY5* could interact with *CmoHY5-1* and *CmoHY5-2,* respectively, to jointly regulate *CmoNRT2.1* expression. The oriental melon scion regulated the nitrate uptake capacity of the melon/squash grafting plant roots, and the higher expression of *CmHY5* in the oriental melon scion leaves, the more substantial the nitrate uptake capacity of squash rootstock roots.

## 1. Introduction

The grafting technique is widely used in plant production to overcome continuous cropping obstacles and soil-borne diseases, increase crop yield, and improve resistance to adversity [1,2,3,4,5,6]. In addition, grafting could effectively improve the plant’s ability to absorb water and nutrients. Studies have shown that grafted tomatoes, apples, and other plants have a more robust nutrient absorption capacity than own-root plants [1,7,8,9,10]. Using squash or gourd as the rootstock, grafting watermelon plants had a more substantial absorption capacity of nitrogen, potassium, calcium, magnesium, iron, and manganese [11]. However, the physiological and molecular mechanism of grafts regulating plant growth, development, and morphogenesis becomes more complicated due to the genomes of two or more species in grafted plants. 

The various small molecules (small RNA, messenger RNA) involved in the interaction of the genome of grafted plants [12,13,14,15,16,17,18,19,20,21] caused a large number of new gene modifications and expression changes in rootstocks and scions, as well as changes in the phenotype and function of grafted plants [22,23]. Small RNAs such as miRNA and siRNA, the long-distance messengers, move systematically and play an essential role in chromatin modification, mRNA stability, and the downstream regulation of translation [24]. miRNA and siRNA could move from cell to cell through vasculature [16], induce gene silencing, inhibit gene expression [25,26,27], and participate in epigenetic modifications in plants [28,29]. mRNAs’ movement in phloem regulated the plant growth and development and defense responses. However, not all mobile mRNAs performed their functions. The mRNAs as regulators of scion–rootstock interaction could be translated into proteins at the transport location. Moreover, mRNAs transported only occurred in specific transcripts [30]. Many mobile mRNAs have been identified using RNA-Seq in multiple species. mRNAs might be transported from stems and leaves to roots and in reverse [31]. The 138 mobile mRNAs were identified in the heterograft combination of arabidopsis and tobacco [32], and a large number of mobile mRNAs were also detected in grafted grapes by transcriptome detection [33]. In the homograft arabidopsis, proteins in the cells associated with the new buds were transported to the root and participated in the regulation of root growth [34]. It was found that transcription factor SlCyp1 could be transported from scion to rootstock in grafted tomato plants and regulate the root–shoot ratio [35].

ELONGATED HYPOCOTYL5 (HY5) is a long-distance transport transcription factor that regulates the expression of genes containing cis-acting elements G-BOX or Z-BOX and miRNAs [36,37,38,39,40]. Notably, the long-distance transport transcription factor HY5 is essential in regulating plant nutrient uptake and accumulation. HY5 could regulate the expression of sulfite reductase genes, which play an essential role in plant sulfate assimilation [41]. HY5 also interacted with copper signal transcription factor SPL7 and affected copper signal allocation by regulating miRNA408 expression [42]. HY5 was also involved in regulating the expression of nitrite reductase genes and ammonium transporter protein genes to affect nitrogen uptake and utilization in plants [43]. HY5 affected nitrate uptake in plants by regulating the expression of nitrate transporter NRT2.1 [44]. Moreover, in the homograft experiments, HY5 could transport long distances from the scion to the root system, playing an essential role in scion–rootstock interaction [44]. However, how long-distance transcription factor HY5 regulates the nitrate uptake capacity of the heterograft plant, such as the grafted plant of melon scion grafted onto the squash rootstock, is even unclear.

In the protected cultivation of melons and watermelon, the squash and pumpkin are often taken as the rootstock. Grafted melons and watermelons not only improve disease and cold resistance but also enhance the absorption capacity of their root system [2,45]. The increased root absorption capacity of grafted melon and watermelon plants is often thought to be due to the rootstock’s well-developed and vigorous roots. However, our team recently found a significant difference in the root absorption capacity of different melon scions grafted onto the same squash rootstocks, but the underlying mechanism has not yet been reported. Therefore, it is theoretically worth clarifying the underlying molecular mechanism of the differences in nitrate uptake of grafted melon plants and further optimizing the selection of rootstocks and scions for grafting cultivation of melon in practice.

## 2. Results

### 2.1. Melon Scions Affect CmoNRT2.1 Expression of Squash Rootstocks to Modulate the Nitrate Uptake Capacity of Grafted Plants

We grafted two different scions of oriental melon (JS-1408 and JS-1429) onto the same squash rootstock (ZM-1507) and obtained two heterologous grafting plants (JS-1408/ZM-1507 and JS-1429/ZM-1507). When the grafted seedlings were grown up with five leaves, we used the ^15^N stable isotope tracer technique to measure and analyze the difference in the nitrate uptake rate and activity between JS-1408/ZM-1507 and JS-1429/ZM-1507. The nitrate uptake rate and activity of JS-1408/ZM1507 were 1.57 and 1.19 times that of JS-1429/ZM1507, respectively (Figure 1A,B). The results showed a significant difference (*p* < 0.001), so we suggested that the significant difference in the nitrate uptake capacity of grafted seedlings was due to the different varieties of oriental melon scions.

The nitrate transporter gene (*NRT)* mainly regulates nitrogen uptake and utilization in plants. The molecular mechanism of nitrate uptake in arabidopsis, rice, tomato, barley, and other crops has been well studied. Plants absorb nitrate from the soil through high-affinity and low-affinity systems, and *NRT2.1*, a member of the nitrate transporter gene families in the high-affinity absorption system, plays an essential role in the absorption of nitrate by roots. Our studies found a significant difference (*p* < 0.01) by measuring the *CmoNRT2.1* expression level in rootstock roots of JS-1408/ZM-1507 and JS-1429/ZM-1507 (Figure 1C). In addition, JS-1408/ZM-1507, with a higher nitrate uptake capacity, also had a higher *CmoNRT2.1* expression level than JS-1408/ZM-1507. Therefore, we hypothesized that the oriental melon scion regulates the expression of *CmoNRT2.1* in the rootstock to control the nitrate uptake capacity of the grafting seedlings. In order to intensely figure it out, we performed a transient silencing assay of *CmoNRT2.1* in the grafted plant roots and measured the nitrate uptake rate and activity using the ^15^N stable isotope tracer technique. The results of the silencing efficiency assay showed that the *CmoNRT2.1* expression of the transient silenced plants was significantly lower than that of the control (Figure 1D). Moreover, the transient silenced plants’ nitrate uptake rate (Figure 1E) and uptake activity (Figure 1F) were significantly lower than those of the control. The results indicated that the oriental melon scion could control the nitrate capacity of the squash rootstocks by regulating *CmoNRT2.1* expression in the grafted plants.

### 2.2. Melon Scion CmHY5 mRNA Undergoes Long-Distance Transport to the Rootstock and Modulates the Nitrate Uptake by Regulating CmoNRT2.1 Expression of Squash Rootstock Roots

Plant roots’ nitrate uptake capacity is closely related to nitric acid transporters (NRTs) and is also affected by light. It has been found that HY5, a long-distance transcription factor, plays an essential regulatory role in plant nitrogen absorption and assimilation [44,46]. HY5 is a bZIP transcription factor that can sense light signals and transmit light signals to photoreceptors, thus regulating plant growth and development [47,48]. In order to clarify whether the different oriental melon scions had a significant effect on the nitrate uptake capacity of grafted plants, we compared the expression level of *CmHY5* in the leaves of own-root grafted seedlings (JS-1408 and JS-1429) using the TaqMan technique and found that the expression level of *CmHY5* in the leaves of JS-1408 was more than two times that of JS-1429 (Figure 2A). Then, we determined the expression level in the scion leaves, scion stems, rootstock stems, and rootstock roots of two kinds of grafted plants (JS-1429/ZM-1507 and JS-1408/ZM-1507). The results showed that *CmHY5* mRNA could be transported over long distances into rootstocks, and the expression level of *CmHY5* in four different parts of JS-1408/ZM-1507 was significantly higher than that of JS-1429/ZM-1507 (Figure 2B–E).

Interestingly, the grafted plants, JS-1408/ZM-1507, with high *CmoNRT2.1* expression levels and a strong nitrate uptake capacity, also had high *CmHY5* expression levels. To examine whether *CmHY5* mRNA was translated and transported into the rootstocks, transient expressions of *CmHY5*-GFP in melon scion leaves were conducted. We found that the GFP signal could be detected in the phloem of the scion, rootstock stems, and rootstock roots after 72 h (Figure 2F,G). These results suggested that *CmHY5* mRNA in melon scion transported into the squash rootstocks might significantly promote nitrate uptake by up-regulating *CmoNRT2.1* expression in grafted plants.

In order to further explore the relationship between transcription factor *CmHY5* and the nitrate uptake capacity of grafted melon plants, we used the VIGS technique to transient silence *CmHY5* in melon scions and measured the transient silencing efficiency (Figure 2H). The results showed that the *CmHY5* expression level was significantly reduced in grafted plants’ rootstock stems and roots (Figure 2I,J), and the relative expression level of *CmoNRT2.1*, the nitrate uptake rate, and the nitrate uptake activity as well (Figure 2K–M). Therefore, it was evidence that transcription factor *CmHY5* was transported into the rootstock from the scions to regulate the nitrate uptake of grafted melon plants.

However, the underlying molecular mechanism of the long-distance transcription factor *CmHY5* regulating nitrate uptake in the oriental melon scion grafted onto the squash rootstock remains unclear. We analyzed the promoter of *CmoNRT2.1* and found it contained the two cis-acting elements G-box that *CmHY5* could bind (Figure 2N). Then, we conducted a yeast single hybridization test between *CmHY5* and the *CmoNRT2.1* promoter. The results showed that *CmHY5* could bind to the promoter of *CmoNRT2.1* (Figure 2O). Since the two binding elements are the same, we designed the hot probe, cold competitive probe, and mutation cold competitive probe using the G-box element and carried out in vitro EMSA, further proving that *CmHY5* could bind to the G-box element on the promoter of *CmoNRT2.1* (Figure 2P). We also performed a GUS reporter gene test in tobacco, and the infected leaves were punched and stained (Figure 2Q). Meanwhile, the activity of the GUS protein was also measured (Figure 2R), indicating that *CmHY5* could be bound to the promoter of *CmoNRT2.1* and positively regulate its expression (*p* < 0.05). We speculated that the transcription factor *CmHY5* might positively regulate the significantly increased expression of *CmoNRT2.1* through the other molecular mechanisms.

### 2.3. CmoHY5 in Rootstocks Modulates the Nitrate Uptake by Regulating CmoNRT2.1 Expression of Squash Rootstock Roots

The above experiments demonstrated that *CmHY5* could bind to the *CmoNRT2.1* promoter and positively regulate its expression. However, *CmoHY5* was present in squash rootstocks in the heterograft plants of oriental melon scion grafted onto squash rootstock, so we suggest that *CmoHY5* of squash rootstocks might affect the *CmoNRT2.1* expression, and we were uncertain of *CmoHY5* like *CmHY5* to long-distance transport. Therefore, we measured the expression of *CmoHY5-1* and *CmoHY5-2* in different parts of JS-1429/ZM-1507 and JS-1408/ZM-1507 (scion leaf, scion stem, rootstock stem, and rootstock root). The results indicated that *CmoHY5-1* and *CmoHY5-2* expressions were not detected in the melon scion leaves and stems, while *CmoHY5-1* and *CmoHY5-2* expressions could be detected in the squash rootstock stems and roots of JS-1429/ZM-1507 and JS-1408/ZM-1507 (Figure 3A,B). Remarkably, the expression of *CmoHY5-1* and *CmoHY5-2* in the stems and roots of JS-1408/ZM-1507 was significantly higher than those of JS-1429/ZM-1507. In addition, the expression of *CmoHY5-1* was significantly higher than that of *CmoHY5-2* in JS-1429/ZM-1507 and JS-1408/ZM-1507. According to the nitrate uptake capacity of JS-1429/ZM-1507 and JS-1408/ZM-1507 (Figure 1A,B), we concluded that the expression of *CmoHY5-1* and *CmoHY5-2* in the squash rootstocks was also positively correlated with the nitrate uptake rate and uptake activity of the grafted melon plant.

Therefore, we speculate that *CmoHY5-1* and *CmoHY5-2* also regulated the expression of *CmoNRT2.1*. Then, the yeast one-hybrid assay demonstrated that both *CmoHY5-1* and *CmoHY5-2* could bind to the promoter of *CmoNRT2.1* (Figure 3C). The EMSA assay also showed that *CmoHY5-1* and *CmoHY5-2* could bind to the G-BOX on the *CmoNRT2.1* promoter (Figure 3D,E). In addition, we also performed the tobacco GUS reporter gene assay and stained the infested leaves by punching, and GUS protein activity was measured. The results indicated that both *CmoHY5-1* and *CmoHY5-2* positively regulated the expression of *CmoNRT2.1*. However, the promoter activity of *CmoHY5-2* on *CmoNRT2.1* was weak compared to the promoter activity of *CmoHY5-1* on *CmoNRT2.1*(Figure 3F–I). Thus, during this process, *CmoHY5-1* has a significant regulatory role in the expression of *CmoNRT2.1*.

### 2.4. CmHY5 in Melon Scion Positively Regulates CmoHY5 Expression in Squash Rootstock

Given the experimental results, *CmHY5*, *CmoHY5-1*, and *CmoHY5-2* all positively regulated the expression of *CmoNRT2.1*. In addition, there were also highly significant differences in the expression of *CmoHY5-1* and *CmoHY5-2* in rootstock stems and the roots of the grafted plants with two different melon scions grafted onto one type of squash rootstock (JS-1429/ZM-1507, JS-1408/ZM-1507) (Figure 3A,B). Furthermore, the grafted plants with the higher expression of *CmHY5* owned the higher expression of *CmoHY5-1* and *CmoHY5-2*. However, the direct promotion of *CmoNRT2.1* expression by *CmHY5* was insignificant, so we speculated that *CmoHY5* mediated the regulation of the squash rootstock *CmoNRT2.1* by the melon scion *CmHY5*. To confirm the relationship between *CmHY5* and *CmoHY5*, we first measured the expression of *CmoHY5-1* and *CmoHY5-2* in the rootstock stems and roots of grafted melon plants with transient silenced CmHY5 in melon scions and found that grafted plants with a lower expression of *CmHY5* had a lower expression of *CmoHY5-1* and *CmoHY5-2* in the rootstock stems and roots (Figure 4A–D). Moreover, we analyzed the promoters of *CmoHY5-1* and *CmoHY5-2* and found that they both contained the *CmHY5* binding element G-BOX in their promoters (Figure 4E). A yeast one-hybrid assay was performed using *CmHY5* with the promoters of *CmoHY5-1* and *CmoHY5-2*, respectively. The results showed that CmHY5 could bind to the promoters of *CmoHY5-1* and *CmoHY5-2*, respectively (Figure 4F). An EMSA assay was also carried out with these two components, and the results showed that *CmHY5* could bind to both components (Figure 4G,H). In addition, we carried out a tobacco GUS reporter gene assay, in which GUS staining and GUS protein activity were measured on infested tobacco leaves. The results demonstrated that *CmHY5* positively regulated the expression of *CmoHY5-1* and *CmoHY5-2*, and the expression of *CmoHY5-1* was more effectively promoted (Figure 4I–L).

### 2.5. CmHY5 Protein of Melon Scion Interacts with CmoHY5 Protein of Squash Rootstock

Previous studies have shown that transcription factor HY5 not only forms dimers on its own to function but also binds to its promoter to accelerate its expression under light [49,50]. Therefore, we suggested that the CmHY5 protein of melon scion might interact with the CmoHY5 protein of squash rootstock. On the one hand, the yeast two-hybrid assay showed that CmHY5 protein could interact with CmoHY5-1 and CmoHY5-2, respectively (Figure 5A). We carried out further fluorophore complementation assays and bimolecular fluorescence complementation assays, which also showed the same results (Figure 5B,C).

## 3. Discussion

Grafting uses the rootstock root to replace the scion root, thus improving the grafting plants’ resistance and nutrient uptake. Many studies have shown that grafting could improve the uptake of mineral nutrition such as nitrogen, potassium, calcium, magnesium, iron, and manganese [7,9,10,11,45,51,52]. In the grafted plant, exchanging information between rootstock and scion is a mutual process. It has been shown that the rootstock controls the accumulation and distribution of salt ions in the scion [53,54], and scion growth and yield are also regulated by the rootstock [55]. In addition, the rootstock affects the biomass of the scion, while the scion also affects the biomass of the rootstock [56,57,58]. However, our studies found that two grafting combinations of the two different varieties of oriental melon scion grafted onto the same squash rootstock showed highly significant differences in nitrate uptake rate and activity (Figure 1A,B). In further investigation, we found that *CmoNRT2.1* regulated the nitrate uptake capacity of squash rootstock roots and the higher the expression of *CmoNRT2.1*, the better the nitrate uptake capacity of the grafted plants (Figure 1C–F). We therefore suggested that the oriental melon scion would have a regulatory effect on the nitrate uptake of the squash rootstock, a process involving long-distance transport and regulation of signaling substances in the grafted plant.

The regulation of scion to rootstock in grafted plants is regulated by various long-distance transport signals such as hormones, RNA, DNA, and proteins [24,59,60,61,62,63]. Studies in apples have shown that more growth hormone is transported downwards when multi-branched apple scions are grafted on the same rootstock compared to controls, resulting in higher growth hormone and cytokinin levels in the roots and suppressed root growth [64]. In addition, gibberellins and cytokinins are mobile and transported in grafted plants and are involved in regulating graft healing and plant growth and development [65,66]. Besides the plant hormones, long-distance transport of RNA, DNA, and proteins occurs in plants, with relatively few processes involving proteins and DNA compared to RNA, and various studies have shown that proteins can accompany mRNA transport [59,67,68]. It can also be long-distance transported to the rootstock in grafted tomato plants. SlCyp1 was transported from the scion to the rootstock and involved in regulating the root-to-shoot ratio [35]. In the arabidopsis homologous grafting combination, proteins from the companion cells of the new shoot can be transported to the root and regulate root growth [34]. DNA interchange provides the basis for graft-induced genetic variation, and grafting provides a channel for forming asexual species [69,70]. In the grafting complex, RNA has a critical role in substances that are transported over long distances, including various types of small RNAs as well as mRNAs [61,62,71,72,73]. It has been shown that mRNA transport is involved in stress signaling and response and fruit quality regulation [74,75], and mobile mRNA CAX3 regulates iron and zinc uptake [76]. It has also been shown that the transport of non-cell autonomous mRNAs is regulated by specific RNA mobile elements and is also related to the structure of the RNA [30]. Therefore, studying the role of mRNAs in rootstock is essential for the future regulation of growth and development, morphogenesis, and fruit quality in grafted plants. In this study, we examined *CmHY5* mRNA in all tissues (oriental melon scion leaves and stems, squash rootstock stems and roots) and found that RNA was detected in all tissues, but the mRNA content of *CmHY5* was lower in the rootstock stems and roots compared to the scion leaves and stems, especially in the rootstock root system, which was 58–89 times lower compared to the scion leaves. The results indicated that *CmHY5* mRNA could be transported as mRNA (Figure 2A–E). In addition, we attached *CmHY5* to GFP fluorescence and then infiltrated the leaves of oriental melon scion in grafted plants (Figure 2F,G). After 72 h, GFP signals were detected in the oriental melon scion stem, squash rootstock stem, and root, further indicating that *CmHY5* of the oriental melon scion could be transported to the squash rootstock, either in the form of protein or possibly translated after transport to function. Several studies have been conducted on HY5 as a signaling substance for long-distance transport. HY5 can function through branches sensing FR light to the root system to regulate root growth [77] and transported from leaves to roots to activate the expression of FER transcription factors and regulate iron uptake in plants [78]. In addition, HY5 can be transported from the silique to the root system to regulate lateral root growth and orientation [77,79,80,81]. We also measured the *CmoHY5* expression in various tissues and showed that no expression of *CmoHY5-1* and *CmoHY5-2* was detected in the oriental melon scions and that expression was higher in both the stems and roots of the squash rootstock, indicating that *CmoHY5* produced by the rootstock is not transported upwards. Furthermore, it has been shown that HY5 can form dimers on its own to function and bind its promoter to promote its expression in the light [49,50]. Therefore, we hypothesized that *CmHY5* produced by the scion would promote *CmoHY5* expression in the rootstock, so we next demonstrated that *CmHY5* produced by the oriental melon scion would indeed positively regulate both *CmoHY5-1* and *CmoHY5-2* in the squash rootstock using a Y1H assay, an EMSA assay, a tobacco GUS reporter gene assay, and a VIGS transient silencing assay (Figure 4). Among them, the promotion of *CmoHY5-1* was highly significant, whereas the promotion of *CmoHY5-2* was insignificant, and our results demonstrated that inter-binding of HY5 could also occur in different species (Figure 5). Moreover, they all positively regulated the *CmoNRT2.1* expression to modulate the nitrate uptake capacity of the grafted plants (Figure 2 and Figure 3).

## 4. Materials and Methods

### 4.1. Plant Materials and Growth Conditions

Oriental melon ‘JS-1408’ and ‘JS-1429’ cultivars (*Cucumis melo* var. *makuwa* Makino) were, respectively, grafted onto squash, ShengZhen No.1 cultivar (*C. moschata*), with the one-cotyledon graft method [82]. Grafted seedlings were transplanted into the nutritional bowl (12 cm × 12 cm) in the glasshouse at Shenyang Agriculture University, Liaoning, China (41°49′ N 123°32′ E). When the grafted plants, JS-1408/ZM-1507 and JS-1429/ZM-1507, grow to five leaves, we took the samples for subsequent qRT-PCR analysis, TaqMan analysis, and ^15^N stable isotope tracer test.

### 4.2. ^15^N Stable Isotope Tracing Assay

When the grafted seedlings grew to five leaves and one heart, each grafted seedling was fed with 20 mL (5 mM) K^15^NO_3_ solution (abundance 10.16%), and the grafted seedlings were divided into aerial parts from the grafting interface until the 48th hour of tracing. Take samples from the underground part, fix the green at 105 °C for 30 min, bake in an oven at 70 °C for 3–5 days to entirely remove moisture, then grind the sample and pass it through a 100-mesh sieve: ratiometric mass spectrometer (Iso Prime100 Isotope Ratio Mass Spectrometer, Germany) determination, three biological replicates. The δ^15^N in the samples was determined by the elementary analysis–isotope ratio mass spectrometers (EA-IRMS) method [83]. The calculation formula for ^15^N absorption rate and absorption activity is as follows:

^15^N absorption rate = (SN + RN) × (^15^N feeding amount)^−1^ × 100%;

^15^N absorption activity = (SN + RN) × (DWR × h)^−1^.

(Note: SN, ^15^N content in rootstock roots; RN, ^15^N content in scion shoots; DWR, dry rootstock root weight; h, hours.)

### 4.3. RNA Extraction and Reverse Transcription

TRIzol RNA isolation reagent (Kangwei Century, Taizhou, China) extracted scion leaves and stems and rootstock stems and roots of two grafting combinations (JS-1408/ZM-1507 and JS-1429/ZM-1507), as well as transiently silenced plants and silenced control plants total RNA from roots. The cDNA was obtained using a reverse transcription kit (TaKaRa, Beijing, China).

### 4.4. qRT-PCR

qRT-PCR was used to detect the silencing efficiency of the transient silencing test and to determine the relative expression of *CmoNRT2.1* in squash rootstock roots. qRT-PCR was performed with a PCR machine (Bio-Rad, CFX96) using the Pro Taq HS SYBR Green premix qPCR kit (AG) in a 20 μL reaction system (2× SYBR Green Pro Taq HS Premix 10 μL, cDNA 2 μL, forward and reverse primer 0.4 μL each, ddH_2_O 7.2μL) was used for qRT-PCR analysis, with thermal cycling as follows: 95 °C for 2 min, 40 cycles for 10 s at 95 °C, and 30 s at 60 °C. 

### 4.5. TaqMan

TaqMan was used to detect the expression levels of *CmHY5*, *CmoHY5-1*, and *CmoHY5-2* in various tissues. It was performed with a PCR machine (Bio-Rad, CFX96) using SuperReal fluorescence quantitative premix reagent (probe method) kit (TIANGEN) in a 20 μL reaction system (2× Super Real Premix (Probe) 10 μL, cDNA 2 μL, forward and reverse 0.6 μL each, primer 0.4 μL, ddH_2_O 6.4 μL)) for TaqMan analysis, with thermal cycling as follows: 95 °C for 15 min, 40 cycles for 3 s at 95 °C, and 25 s at 60 °C. The probe was modified by 5’-FAM. The Ct values obtained by the analysis were substituted into the standard curves of the three genes to calculate the absolute expression levels. The standard curves were made as follows: the mRNA sequences of *CmHY5*, *CmoHY5-1*, and *CmoHY5-2* were connected to the TA cloning vector, and the recombinant plasmid was amplified by 10-fold gradient dilution. The diluted plasmid was used as a standard using the above system and method. TaqMan assay was performed, and a standard curve was drawn based on the natural logarithm of the Ct values and the plasmid concentrations. The standard curves of *CmHY5*, *CmoHY5-1*, and *CmoHY5-2* were prepared, respectively. 

### 4.6. Virus-Induced Gene Silencing (VIGS)

The transient silencing test mediated by tobacco ringspot virus was used [84], the coding sequence (CDS) sequence of *CmHY5* (477 bp) was inserted into the *Sna*BI restriction site of the pTRSV2 RNA2S vector. Then, the pTRSV2-CmHY5, P19, pTRSV RNA1, and pTRSV2-CuPDS were transformed into Agrobacterium-competent EHA105. P19, pTRSV RNA1, and pTRSV2-CmHY5were suspended to OD_600_ = 1.0 and injected into *N. benthamiana* leaves at a ratio of 1:1:1. pTRSV RNA1+ pTRSV2-CuPDS+P19 was used as positive disease control, and pTRSV RNA1+ pTRSV2+P19 was used as a silent control. After the diseased tobacco leaves were collected, the diseased leaves were collected for friction inoculation of fully healed thin-skinned muskmelon/Chinese pumpkin grafted scion leaves. Ten to fifteen days for sampling and ^15^N stable isotope tracer assay.

### 4.7. Subcellular Localization of CmHY5, CmoHY5-1, CmoHY5-2, and CmoNRT2.1

Full-length CDs of *CmHY5*, *CmoHY5-1*, *CmoHY5-2*, and *CmoNRT2.1* were cloned into the 35S::GFP vector. Subsequently, the recombinant vectors were transformed into Agrobacterium-competent EHA105 and injected into *N. benthamiana* leaves. The leaves were stained with DAPI for 5 min to clarify *CmHY5*, *CmoHY5-1*, and *CmoHY5-2* subcellular localization. As previously described [85], confocal imaging was performed with an FV3000 microscope (Olympus). 

### 4.8. Y1H Assay

Full-length CDs of *CmHY5*, *CmoHY5-1*, and *CmoHY5-2* were cloned into the pGADT7 vector to generate prey. The full-length promoter of *CmoNRT2.1* and four promoter fragments, Pro1 (291 bp with two G-box), Pro2 (228 bp with a G-box), Pro3 (217 bp with a G-box), and Pro4 (111 bp with a G-box) of *CmoHY5-1* and *CmoHY5-2* were amplified and cloned into the pAbAi vector, producing five baits. All bait vectors require linearization and purification before transforming yeast Y1H-Gold. The bait vector was transformed into yeast Y1H-Gold separately and plated on SD/-Ura +50/100/150/200/300 ng/mL Aureobasidin A (AbA) medium and incubated at 30 °C for 3 days to screen AbA concentration. The prey vector pGADT7-*CmHY5* was transformed into yeast with baits vector pAbAi-Pro1, pAbAi-Pro2, pAbAi-Pro3, and pAbAi-Pro4, separately. The prey vectors pGADT7-*CmHY5*, pGADT7-*CmoHY5-1*, and pGADT7-*CmoHY5-2* were transformed into yeast with baits vector pAbAi-Pro*CmoNRT2.1,* separately, along with positive (pGAD-p53+p53-AbAi) and negative (pGADT7-AD+bait-pAbAi) controls, were plated on SD /-Leu+150 ng/mL AbA and incubated at 30 °C for 3 days. 

### 4.9. Electrophoretic Mobility Shift Assay (EMSA)

Full-length CDs of *CmHY5*, *CmoHY5-1,* and *CmoHY5-2* were cloned into the pD2P_1.06eTM vector. Standard Mini Kit(Kangmaxin Intelligent Technology Co., Ltd. Shanghai, China) was used for protein induction following the manufacturer’s instructions. EMSA was performed using the Light Shift Chemiluminescent EMSA Kit (Beyotime, Shanghai, China), following the manufacturer’s instructions. Probes synthesized from yeast one-hybrid results. Two oligonucleotide probes with G-box were synthesized based on the sequences of the *CmoHY5-1* promoter and *CmoHY5-2* promoter, and one oligonucleotide probe with G-box was synthesized based on the sequences of the promoter of *CmoNRT2.1* and biotin-labeled by Saibaisheng (Beijing, China). The unlabeled probes were used as competition probes. The unlabeled mutant probe was used as a mutant cold competition probe. The fusion protein was incubated at 25 °C with the probes, with or without the competition probe and mutant probe, in a total reaction volume of 10 µL. Protein–DNA complexes were separated on a 6% polyacrylamide gel and electroblotted onto a Hybondnylon membrane (Biosharp, Shanghai, China), which was then subjected to UV crosslinking and chemiluminescence detection. 

### 4.10. GUS Assay

Full-length CDs of *CmHY5*, *CmoHY5-1*, and *CmoHY5-2* were cloned into the pRI101 vector to generate an effector vector. Promoter fragments with G-box of *CmoNRT2.1*, *CmoHY5-1*, and *CmoHY5-2* were amplified and cloned into the pBI101 vector to generate the reporter vector. The recombinant vectors were transformed into Agrobacterium-competent EHA105. The reporter and effector co-infect *N. benthamiana* leaves, and GUS activity and staining measurements were performed as previously described [64,86], along with the negative (pRI101+ reporter) controls. 

### 4.11. Y2H Assay

Full-length CDs of *CmHY5* were cloned into the pGBKT7 vector to generate bait. Full-length CDs of *CmoHY5-1* and *CmoHY5-2* were cloned into the pGBKT7 vector to generate prey. Recombinant plasmids were introduced into yeast strain Y2H-Gold, and the transformed yeast cells were plated on SD/-Trp/-Leu medium and incubated at 30 °C for 3 days. Then, the colonies were transferred to SD/-Trp/-Leu/-Ade/-His/AbA150/X-α-Gal medium incubated at 30 °C for 3 days. 

### 4.12. Luciferase Complementation (LCA) Assay

Full-length CDs of *CmHY5* were cloned into the NLuc vector. Full-length CDs of *CmoHY5-1* and *CmoHY5-2* were cloned into the CLuc vector. Recombinant plasmids were introduced into Agrobacterium-competent EHA105 (pSoup) and expressed in tobacco leaves. Fluorescence was observed using an imaging system (Berthold, Lb985, Bad Wildbad, Germany). 

### 4.13. Bimolecular Fluorescence Complementation (BiFC) Assay

Full-length CDs of *CmHY5* were cloned into the pCAMBIA1300-35s-N-YFPC vector. Full-length CDs of *CmoHY5-1* and *CmoHY5-2* were cloned into the pCAMBIA1300-35s-C-YFPC vector. Recombinant plasmids were introduced into Agrobacterium-competent EHA105 (pSoup) and expressed in tobacco leaves. The leaves were stained with DAPI for 5 min, and confocal imaging was performed with an FV3000 microscope (Olympus) as previously described [85]. 

## 5. Conclusions

In conclusion, the *CmHY5* expression level of the scion played a vital role in the nitrate uptake capacity of oriental melon scion grafted on squash rootstock. The scion-to-rootstock mobile transcription factor *CmHY5* induced the *CmoHY5-1* and *CmoHY5-2* expression of squash rootstock and interacted with each other. Moreover, they positively regulated *CmoNRT2.1* expression jointly and individually to modulate the nitrate uptake of grafted plants (Figure 6). The grafted plants with the higher *CmHY5* mRNA level of the oriental melon scion possessed a stronger nitrate uptake capacity.

## Figures and Tables

**Figure 1 ijms-24-00162-f001:**
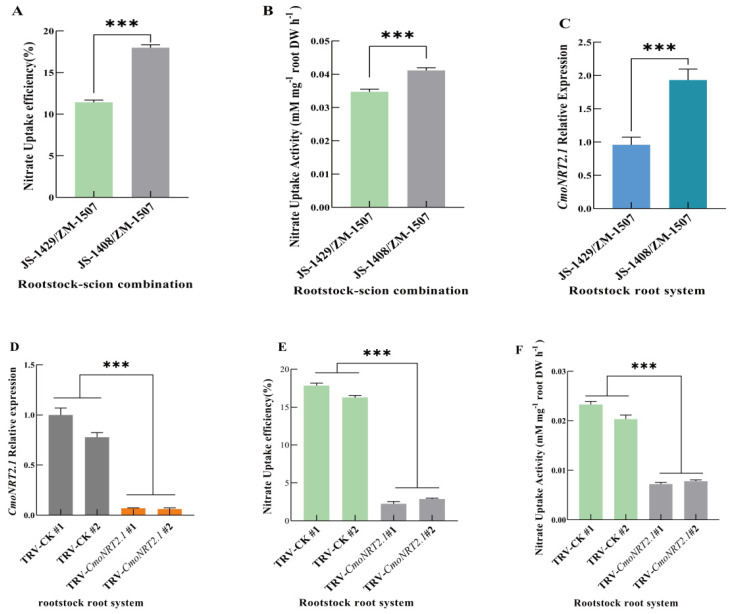
Analysis of differences in nitrate uptake capacity and *CmoNRT2.1* expression of grafted melon plants with two different varieties of oriental melon scions grafted onto the same squash rootstocks during the five leaves growth stage. *CmoNRT2.1* transiently silenced plants: TRV-*CmoNRT2.1*; Control plants: TRV-CK. Values are means ± SD, *n* = 3 (biological replicates). Asterisks indicate statistically significant differences (*** *p* < 0.001, Student’s t-test). (**A**,**B**). Nitrate uptake efficiency and nitrate uptake activity by ^15^N stable isotope tracing assay. (**C**). Quantitative real-time (qRT) PCR analysis of *CmoNRT2.1* expression. (**D**). Analysis of the silencing efficiency of transient silenced *CmoNRT2.1* in the grafted plants. (**E**,**F**). Analysis of nitrate uptake rate and uptake activity in the grafted plants’ roots system of transient silenced *CmoNRT2.1*.

**Figure 2 ijms-24-00162-f002:**
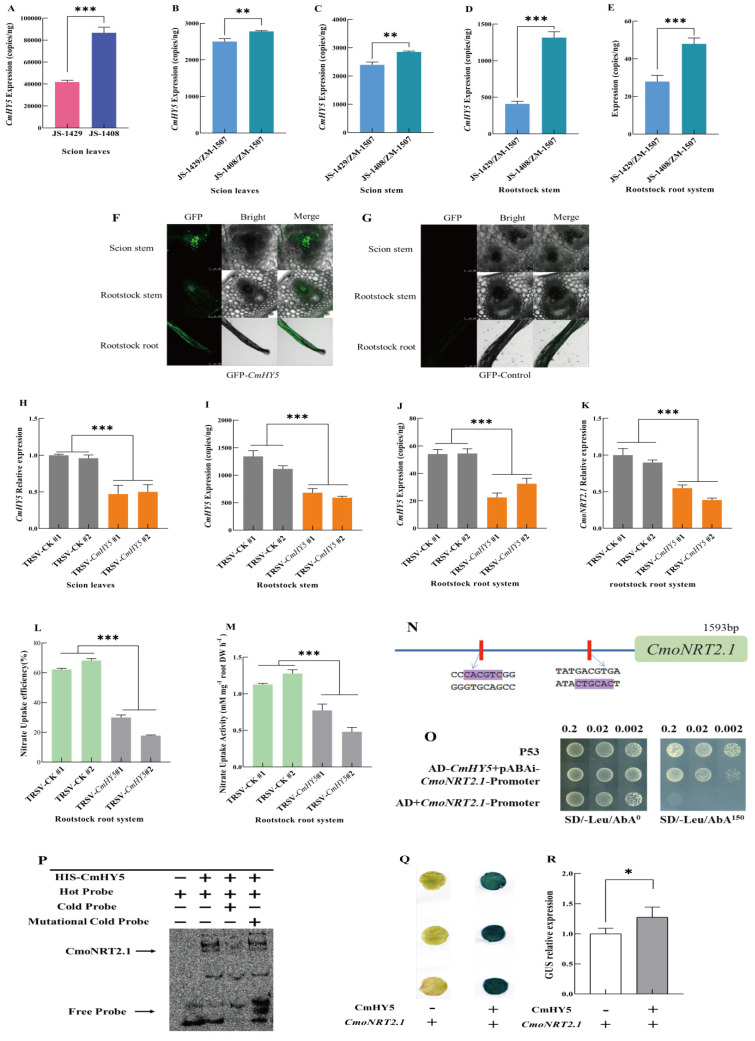
*CmHY5* in the oriental melon scion positively modulates the nitrate uptake of squash rootstock roots by regulating *CmoNRT2.1* expression. Values are means ± SD, *n* = 3 (biological replicates). Asterisks indicate statistically significant differences (*** *p* < 0.001, ** *p* < 0.01, * *p* < 0.05, Student’s t-test). (**A**). TaqMan analysis of *CmHY5* expression in the leaves of two different own-grafted melon plants. (**B**). TaqMan analysis of *CmHY5* expression in the scion leaves of grafted plants with two different varieties of oriental melon scions grafted onto the same squash rootstock. (**C**). TaqMan analysis of *CmHY5* expression in the scion stems of grafted plants with two different varieties of oriental melon scions grafted onto the same squash rootstock. (**D**). TaqMan analysis of *CmHY5* expression in the rootstock stems of grafted plants with two different varieties of oriental melon scions grafted onto the same squash rootstock. (**E**). TaqMan analysis of *CmHY5* expression in the rootstock roots of grafted plants with two different varieties of oriental melon scions grafted onto the same squash rootstock. (**F**). Analysis of GFP-*CmHY5* transport in the grafted plants. (**G**). Analysis of GFP-Control transport in the grafted plants. (**H**). qRT-PCR analysis of the relative expression of *CmHY5* in the oriental melon scion leaves of grafted plants with transient silenced *CmHY5*. (**I**). TaqMan analysis of *CmHY5* expression in the squash rootstock stems of grafted plants with transient silenced CmHY5. (**J**). TaqMan analysis for detecting *CmHY5* expression in the squash rootstock roots of grafted plants with transient silenced *CmHY5*. (**K**). qRT-PCR analysis of the relative expression of *CmoNRT2.1* in the squash rootstocks of grafted plants with transient silenced *CmHY5*. (**L**,**M**). The nitrogen uptake rate and uptake activity assay of grafted plants with transient silenced *CmHY5*. (**N**). Analysis of the cis-acting element of the *CmoNRT2.1* promoter. (**O**). Yeast one-hybrid assay to detect *CmHY5* binding to the *CmoNRT2.1* promoter. (**P**). EMSA assay to detect *CmHY5* binding to the *CmoNRT2.1* promoter. (**Q**,**R**). GUS staining and GUS activity assay in *N. benthamiana* leaves after transient co-expression of 35S::CmHY5 with pro*CmoNRT2.1*::GUS and co-expression with pBI101:GUS as a control.

**Figure 3 ijms-24-00162-f003:**
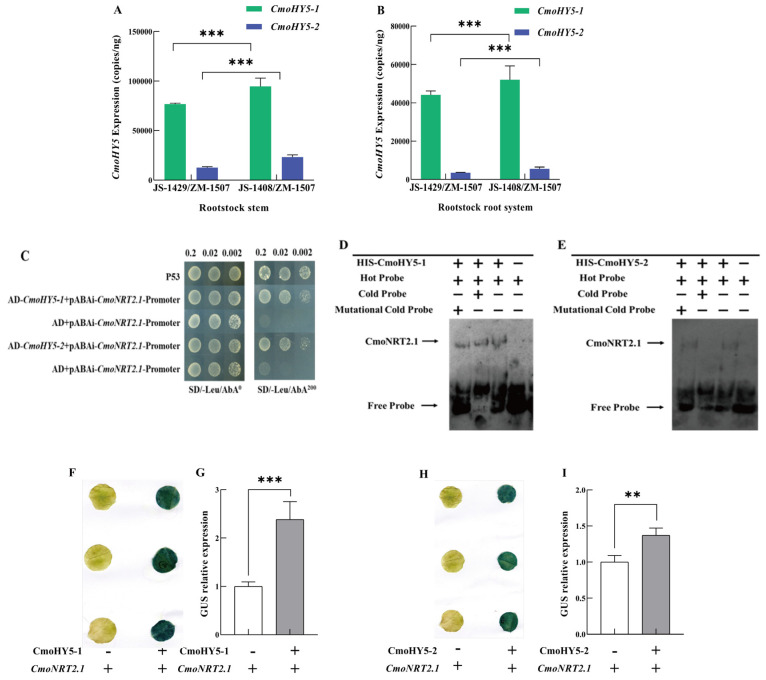
*CmoHY5* regulates the expression of *CmoNRT2.1* in the squash rootstock roots. Values are means ± SD, n = 3 (biological replicates). Asterisks indicate statistically significant differences (*** *p* < 0.001, ** *p* < 0.01, Student’s t-test). (**A**). TaqMan analysis of *CmoHY5* expression in the squash rootstock stems of grafted plants with two different varieties of oriental melon scions grafted onto the same squash rootstock. (**B**). TaqMan analysis of *CmoHY5* expression in the rootstock roots of grafted plants with two different varieties of oriental melon scions grafted onto the same squash rootstock. (**C**). Yeast one-hybrid assay detecting *CmoHY5* binding to the *CmoNRT2.1* promoter. (**D**). EMSA assay to detect *CmoHY5-1* binding to the *CmoNRT2.1* promoter. (**E**). EMSA assay to detect *CmoHY5-2* binding to the *CmoNRT2.1* promoter. (**F**,**G**). GUS staining and GUS activity assay in *N. benthamiana* leaves after transient co-expression of 35S::CmoHY5-1 with pro*CmoNRT2.1*::GUS and co-expression with pBI101:GUS as a control. (**H**,**I**). GUS staining and GUS activity assay in *N. benthamiana* leaves after transient co-expression of 35S::CmoHY5-2 with pro*CmoNRT2.1*::GUS and co-expression with pBI101:GUS as a control.

**Figure 4 ijms-24-00162-f004:**
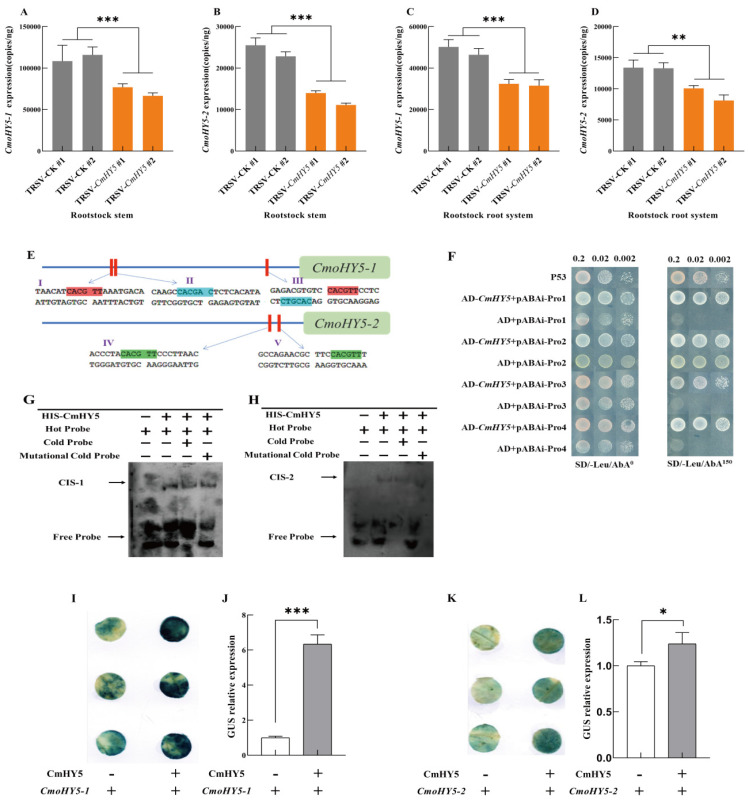
*CmHY5* in the oriental melon scion regulates the expression of *CmoHY5* in the squash rootstock. Values are means ± SD, *n* = 3 (biological replicates). Asterisks indicate statistically significant differences (*** *p* < 0.001, ** *p* < 0.01, * *p* < 0.05, Student’s *t*-test). (**A**). TaqMan analysis of *CmoHY5-1* expression in the squash rootstock stems of grafted oriental melon plants with transient silenced *CmHY5* in the oriental melon scions. (**B**). TaqMan analysis of *CmoHY5-2* expression in the squash rootstock stems of grafted melon plants with transient silenced *CmHY5* in the oriental melon scions. (**C**). TaqMan analysis of *CmoHY5-1* expression in the squash rootstock roots of grafted oriental melon plants with transient silenced *CmHY5* in the oriental melon scions. (**D**). TaqMan analysis of *CmoHY5-2* expression in the squash rootstock roots of grafted plants with transient silenced *CmHY5* in the oriental melon scions. (**E**). Analysis of the promoter cis-acting elements of *CmoHY5-1* and *CmoHY5-2*. (**F**). Yeast one-hybrid assay to detect *CmHY5* binding to the promoters of *CmoHY5-1* and *CmoHY5-2*, respectively. (**G**). EMSA assay to detect promoter binding of *CmHY5* to *CmoHY5-1*. (**H**). EMSA assay to detect promoter binding of *CmHY5* to *CmoHY5-2*. (**I**,**J**). GUS staining and GUS activity assay in *N. benthamiana* leaves after transient co-expression of 35S::CmHY5 with pro*CmoHY5-1*::GUS and co-expression with pBI101:GUS as a control. (**K**,**L**). GUS staining and GUS activity assay in *N. benthamiana* leaves after transient co-expression of 35S::CmHY5 with pro*CmoHY5-2*::GUS and co-expression with pBI101:GUS as a control.

**Figure 5 ijms-24-00162-f005:**
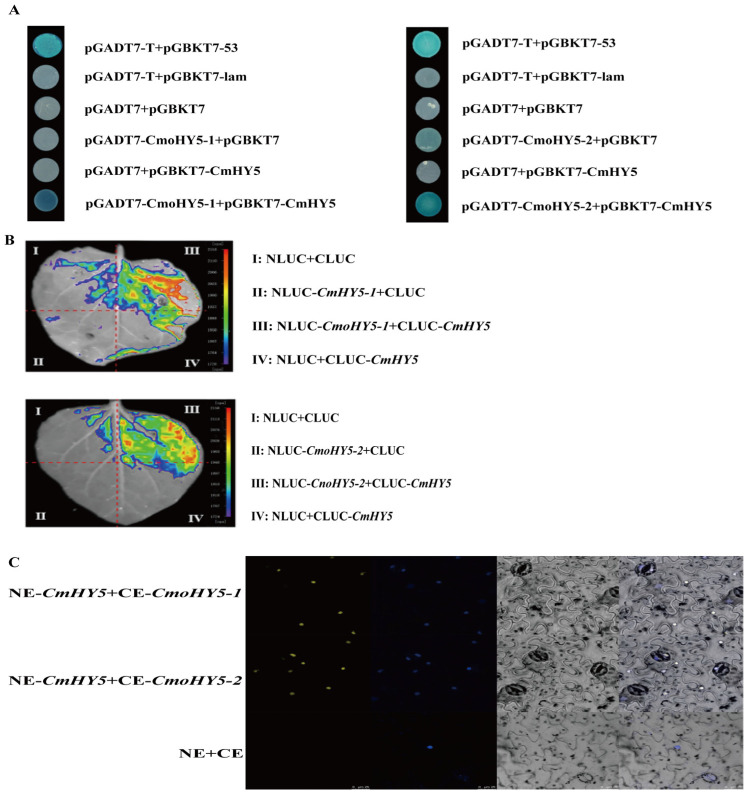
Analysis of protein interaction between CmHY5 and CmoHY5. (**A**). Y2H assay analysis of CmHY5 interacting with CmoHY5-1 and CmoHY5-2 proteins, respectively. (**B**). Fluorophore enzyme complementation assay analysis of CmHY5 interacting with CmoHY5-1 and CmoHY5-2 proteins, respectively. (**C**). Bimolecular fluorescence complementation assay analysis of CmHY5 interacting with CmoHY5-1 and CmoHY5-2 proteins, respectively.

**Figure 6 ijms-24-00162-f006:**
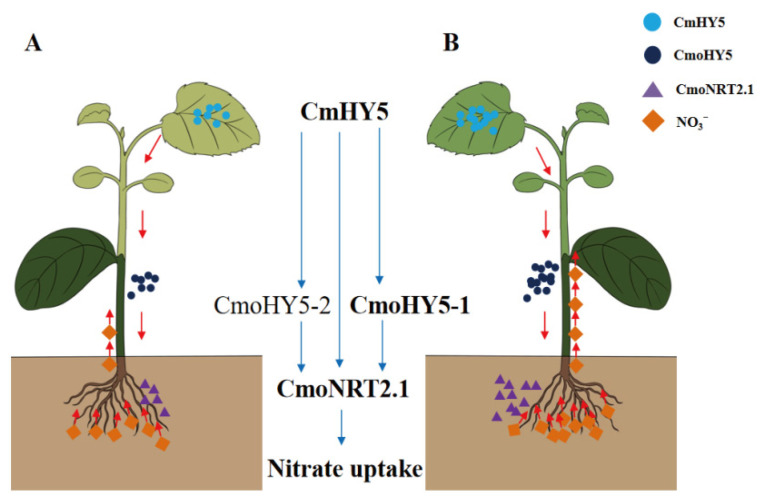
The model of scion-to-rootstock mobile transcription factor *CmHY5* positively modulates the nitrate uptake capacity of oriental melon scion grafted onto squash rootstock. (**A**). The grafted plants of lower *CmHY5* mRNA level. (**B**). The grafted plant of higher *CmHY5* mRNA level.

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
