# Peer review of "Scion-to-Rootstock Mobile Transcription Factor CmHY5 Positively Modulates the Nitrate Uptake Capacity of Melon Scion Grafted on Squash Rootstock"

_ijms, 2022, doi:10.3390/ijms24010162_

Round 1

Reviewer 1 Report

The authors estimated the nitrate uptake capacity of grafted plants in different melon seedling grafted into the same squash root stocks. Some meaning results were presented on this paper. I think it should be published on this journal after revise, but there are still some questions should be improved.

Suggestion 1

1: The language is well organized, but there are still some minor writing errors, such as the arabidopsis hould be Arabidopsis lin Line104. Line431: SnaBI should be SnaBI.

2: For better recognize, all the letters in figures need to be uniform size. Figure 2P and Figure3D: has been irregular impressed and low resolution. Fig4 and Fig 5: many figs are low resolution, It is hard to recognize.

3: Fig3: Why pro:: CmoNRT2.1 have no gus staining individually?

4: There should be minor space between individual pictures in one photo.

5: All the experiments in vitro can prove CmHY5 can promote the expression of CmoHY5-1 and CmoHY5-2. Also, CmoHY5-1 and CmoHY5-2 positively regulated CmoNRT2.1’s expression. Can author suggested some experiments to prove that it can promote the nitrogen absorption and growth under low nitrogen condition. Such, analysis the growth and nitrogen content in the grafted seedling and rootstoch in JS-1408JS1429 JS-1408/ZM-1507JS1429/ M-1507 plants under low and normal nitrogen?

Author Response

Thanks for the reviewers for their generous comments about the writing, figures’ quality and so on. We had made some revisions according to the reviews’ comments in the manuscript.

Reviewer 2 Report

The manuscript by Hou et al. entitled “Scion-to-rootstock mobile transcription factor CmHY5 positively modulates the nitrate uptake capacity of melon scion grafted on squash rootstock” discusses the role of transcription factor CmHY5 present in different oriental melon scions and two other transcription factors (CmoHY5-1 and CmoHY5-2; present in identical squash rootstock), and how they regulate (and interact in) the nitrate uptake (CmoNRT2.1 expression in squash rootstock).

Following are my comments and suggestion to the authors:

Line 44: Write the full form for the abbreviation “mRNA”.

Line 50:  Change “vascular” to “vasculature”.

Line 62: Mention whether ‘SlCyp1’ is a transcription factor.

Line 79: Please omit this sentence. This looks like Line 35.

Line 97/98: Could you possibly put few sentences (with relevant literature) on nitrate uptake rate and activity in the ‘Introduction’ section too?

Line 98 to 102: Please move Line 99-100 right after the sentence ending in Line 98.  Combine sentences (Line 98-99 and Line 101-102) that mention about the significant differences.

Figure 1: Mention the full form of TRV and TRV-CK in the caption or somewhere in the narrative.

Line 133: Is this NRT different than NRT mentioned in Line 103?

Line 134: Is this transcription factor (HY5) different than in Line 64? Please ensure consistent full names of the abbreviations elsewhere in the manuscript.

Line 206: Change ‘presents’ to ‘present’.

Line 206 to Line 211: Please paraphrase the sentences to enhance clarity. I understand that CmoHY5 is a different transcription factor than CmHY5 (present in melon scion). CmoHY5 is present in squash rootstock and has two variants: CmoHY5-1 and CmoHY5-2.

Line 390 to Line 393: Reformat the sentences in the past tense.

Line 394: Write full form for the abbreviation “EA-IRMS”.

Line 401, Line 404: I would suggest adding the name of city and country of the manufacturer in the parentheses here as well as elsewhere in the manuscript.

Line 431: This is the first instance where this abbreviation has been used in the manuscript. Change it to “… the coding sequence (CDS)…”.

Line 431 to Line 433: Please split the sentences for enhanced clarity.

Line 434: Italicize scientific name. Same applies to Line 444, Line 484.

Line 463: Expand the abbreviation ‘EMSA’.

Line 498: Is it ‘pSoup’ plasmid rather than ‘psoup’? Same suggestion applies to Line 505.

Author Response

Thanks for the reviewers for their generous comments and suggestions about the writing and so on. We had made some revisions according to the reviews’ comments and suggestions in the manuscript.
